# Explaining Confident Black-Box Predictions

**Evan Yao**                                                                    *evanyao@mit.edu*
*Operations Research Center*
*Massachusetts Institute of Technology*

**Retsef Levi**                                                                   *retsef@mit.edu*
*Sloan School of Management*
*Massachusetts Institute of Technology*

**Assaf Avrahami**                                                              *aa@wizsoft.com*
*Wizsoft*

**Abraham Meidan**                                                       *abraham@wizsoft.com*
*Wizsoft*

**Reviewed on OpenReview:** *https://openreview.net/forum?id=SAwZpgKJcc*

## Abstract

Interpretability is crucial for leveraging predictive machine learning for decision-making, but the strongest performing models are often black-boxes in that they are difficult to understand. For binary classification models, a growing body of literature seeks to find *model-agnostic* explanations by treating a model as a list of 0/1 predictions and identifying patterns for when a model predicts 1 over 0 (or vice versa). While such explanations are useful for understanding when a model predicts 1 over 0, they do not consider the confidence (i.e., the probability) behind predictions, a critical piece of information provided by most classification models. Since the 0/1 predictions of a model depend on the choice of a subjective threshold for discretizing predicted probabilities, as one changes the threshold, the resulting explanations may change despite the underlying model staying the same. In contrast, this work proposes model-agnostic explanations that treat a black-box model as a *ranking* across a dataset from lowest predicted probability of 1 to highest, rather than a list of 0/1 predictions. Under this ranking, a useful explanation should capture broadly when a model *confidently* predicts 1 (i.e., highly ranked data points). Since highly confident predictions are often correlated with predictions that are more accurate and actionable, understanding when a model predicts confidently is often quite valuable to a practitioner.

This work builds explanations based on rule lists (i.e., a collection of if-then rules) as well as a novel special case called checklists. A strong rule list or checklist is satisfied by a large number of data points that are ranked highly by the model. This criteria is measured by the traditional metric of support (i.e., the number of data points an explanation applies to), the *average* ranking of those data points, which we call the Average Black-Box Ranking (ABBR), as well as the sparsity of the explanation (e.g., number of rules in the rule list, among others). Given these metrics, this work develops a local-search based optimization methodology for finding explanations based on rule lists and checklists that maximize ABBR for a user-specified support and sparsity constraint. The methodology leverages a local search approach where an initial rule list is chosen greedily from a pool of candidate rules, then slowly perturbed by swapping rules from the rule list with those in the candidate pool. This approach is evaluated on 6 real world datasets in application areas ranging from healthcare to criminal justice and finance. Empirical results suggest that this methodology finds rule lists of length at most 5 with ABBR within 7.4% of the optimal ABBR of any explanation, while checklists provide greater interpretability for a small cost in performance.

# 1 Introduction

As digitization increases the amount of available data, an increasing number of human decisions are aided by predictive machine learning models. However, algorithms with the strongest predictive performance are often also the least interpretable, for example random forests Breiman (2001) and neural networks Goodfellow et al. (2016). A growing body of literature seeks to understand such black-box models through *model-agnostic explanations*. For binary classification problems, such approaches treat a model as simply a list of 0/1 predictions (i.e., model-agnostic), and discover patterns that elucidate when the model predicts 1 over 0 (or vice versa). One common type of explanation involves finding simple if-then rules that are (1) *consistent* with a model's binary predictions and (2) apply to large number of instances in the dataset (i.e., high *support*). For example, for a model predicting whether a patient will develop diabetes, consider the following if-then rule:

*If BMI > 32 and High Blood Pressure = 1, then predict* 1 *to diabetes*

This rule provides a useful explanation of when the model predicts 1 by identifying a sizable subset of the population (i.e., those with BMI > 32 and high blood pressure), most of whom the black-box model also predicts 1 (i.e., the rule's prediction of 1 is consistent with the model's prediction). Broadly speaking, an explanation is a statement about the model's prediction on an interpretable subset of the dataset.

While such model-agnostic explanations are useful in understanding when and why a model predicts 1 (or 0), they suffer from at least two drawbacks. First, since most classification models output a list of probabilities, the consistency of such explanations with the black-box model varies greatly depending on a 'subjective' threshold used to convert the probability into 0/1 predictions. Related to this first drawback, the second drawback is that considering only 0/1 predictions, existing approaches overlook the heterogeneity in *confidence* across predictions. A prediction of 1 can be "confident" (i.e., high probability) or just a relatively weak guess, but existing approaches do not distinguish between these two cases. For a well-trained model, the confidence of its predictions is correlated with the model's accuracy, and thus provides valuable information on when the model is most reliable in practice. Arguably, it is more useful for practitioners to understand why a model makes confident predictions over weak predictions as confident predictions are more actionable. For example, a patient strongly predicted to develop diabetes might be put on preventative treatment, whereas a patient with a slightly elevated risk does not require immediate action.

Given these shortcomings, this paper develops model-agnostic explanations that treat a black-box model as a *ranking* of probabilities across the dataset, rather than a discrete list of 0/1 predictions. To the best of our knowledge, this is the first work that provides model-agnostic explanations of the probabilities predicted by black-box models. Our work makes two main contributions.

First, this work proposes a natural way of evaluating the quality of an explanation with respect to a ranking. Explanations are evaluated based on the size of the data points that satisfy the explanation (i.e., support) and the average ranking of those data points, a novel metric called the Average Black-Box Ranking (ABBR). Explanations that *maximize* ABBR provides interpretability for when a black-box model predicts 1 with high confidence. Naturally, one can simply flip the class labels 0 and 1 to find similar explanations for the 0 class. The ABBR metric can be justified through a natural connection to the traditional consistency metric used to evaluate an explanation on discrete 0/1 predictions. While any probability ranking can be discretized into 0/1 predictions by choosing a threshold, an explanation with high ABBR is highly consistent with a model's predictions of 1, averaged across all possible thresholds.

Second, given the ABBR metric, this work develops algorithms `RuleListSearch` and `ChecklistSearch` which seek to find explanations based on *rule lists* and *checklists* respectively. Both algorithms attempt to find explanations with high ABBR given a desired target support. A rule list Angelino et al. (2017; 2018); Letham et al. (2015) is the union (logical OR) of one or more if-then rules; a data point satisfies a rule list if it satisfies any of the rules in it. This work also proposes explanations using checklists, which as the name suggests is a collection of $K$ conditions and a threshold $t$ such that a data point satisfies the checklist if it satisfies at least $t$ conditions. Figure 1a shows an example of a rule list with $M = 4$ and Figure 1b shows a checklist with $K = 4$ and $t = 3$. Both algorithms search for explanations based on local search. Start by generating a large pool of rules or conditions, filtering them down to rules with high ABBR, and searching

| If *any* of the following rules are satisfied:
    (Age > 50 and BMI > 35)
    (BMI > 30 and High Blood Pressure)
    (Heart Disease and BMI > 28 and Smoker)
    (Age > 65 and BMI > 28 and Exercise < 20)
then predict 1 for diabetes. | If at least 3 of the following conditions are satisfied:
  -  Age > 60
  -  BMI > 30
  -  High Blood Pressure
  -  Exercise < 20
then predict 1 for diabetes. |
|---|---|
| a Rule List | b Checklist |

Figure 1: Examples of a Rule List and a Checklist

the space of rule lists or checklists by randomly making swaps with the pool using simulated annealing Kirkpatrick et al. (1983). Both algorithms are also versatile in that a user can control the sparsity of resulting explanation. For rule lists, one can set the number of rules $M$ and the maximum number of conditions in each rule, while for checklists one can set the number of conditions $K$. When evaluated on 6 real-world datasets, `RuleListSearch` generates rule lists consisting of up to 5 rules whose ABBR is within 8% of the best possible ABBR, while checklist rules perform slightly worse, but in return provide a higher degree of interpretability.

**Reproducibility.** Code and datasets will be released on GitHub upon acceptance. Pseudo-code is available in Appendix C.

## 2 Related Work

**Interpretable Rule-Based Classifiers.** The simplest logical building block of any classifier are if-then association rules. Such rules serve as the basis for many interpretable classification algorithms, including decision trees Quinlan (1986); Breiman et al. (2017); Safavian & Landgrebe (1991), rule lists (i.e., an ordered list of if-then statements where earlier rules are prioritized) Liu et al. (1998); Li et al. (2001); Wang & Rudin (2015); Chen & Rudin (2018); Pan et al. (2020); Dash et al. (2018); Letham et al. (2015) or rule sets (i.e, an unordered set of rules where predictions are made by a weighting scheme) Angelino et al. (2017; 2018); Lakkaraju et al. (2016). Such rule based classifiers are often constructed by first mining for a large pool of association rules using association rule-mining algorithms such as Apriori Agrawal et al. (1996) or FPGrowth Han et al. (2004), or using commercially available software like WizWhy Meidan (2005). While such classifiers are highly interpretable, optimizing for them can be challenging due to the discrete nature of association rules.

**Explanations of Black-Box Models.** Recognizing the importance of interpretabilty in practice, a growing body of literature develops techniques for explaining black-box models. One type of explanation is a local explanation, which takes a particular prediction and seeks to understand how small perturbations affect the prediction. This can be done by approximating the prediction locally with a linear model Ribeiro et al. (2016); Yoon et al. (2019); Shankaranarayana & Runje (2019); Zafar & Khan (2021) or classification trees Guidotti et al. (2018). Such explanations are useful in understanding counterfactuals (e.g., what is the best way for a patient to reduce their risk of diabetes). On the other hand, global explanations seek to find overall patterns in how the model predicts, often by approximating the model with an interpretable surrogate Guidotti et al. (2022); Lakkaraju et al. (2016; 2019). Global explanations also cover visualizations for understanding feature importance such as partial dependence plots (PDP) Friedman (2001) or individual continuous expectation (ICE) Goldstein et al. (2015).

Some work also straddles the line between local and global. Shapley scores (SHAP) Lundberg & Lee (2017) can be assigned to an individual prediction, detailing the contribution of each feature to that prediction, but can also be viewed in aggregate across the whole dataset. Recent work on globally consistent rule-based explanations Rudin & Shaposhnik (2023) seek to explain a local prediction by finding a rule with the largest support that is perfectly consistent with the black-box model's 0/1 predictions. An interesting way of adding interpretability to a black-box model is through a 2-stage classifier, where rules are first used to make some predictions, then a black-box model is used for the rest Wang & Lin (2021); Ferry et al. (2023). Such research has shown that part of a black-box model can often be replaced by an interpretable model at little cost to performance.

## 3 Problem Setup and Notation

Denote the input data as $(X_n, y_n)_{n=1}^N$ where $X_n \in \mathbb{R}^D$ is a $D$-dimensional feature vector and $y_n \in \{0, 1\}$ is the target outcome of interest. Let $b : \mathbb{R}^D \to [0, 1]$ be a binary classification model that assigns a predicted probability that the target is 1 for any data point in the feature space. No assumptions are made on the model that yielded these probabilities. Without loss of generality, assume that $(X_n)_{n=1}^N$ in descending order based on $b$, i.e.,

$$b(X_N) \geq b(X_{N-1}) \ldots \geq b(X_2) \geq b(X_1)$$

(ties are broken arbitrarily). For any data point $X_n$, we will refer to $n$ as the *rank* of $X_n$ with respect to $b$.

### 3.1 Rule Lists and Checklist Rules

We now define rule lists and checklists, the structures of the explanations studied in this paper. To build up to defining a rule list, we first define a condition and a rule.

#### 3.1.1 Conditions.

A condition $c = (d, o, v)$ consists of a feature $d \in [D]$, an operator $o \in \{\leq, =, \geq\}$ and a value $v \in \mathbb{R}$, for example "age $\geq$ 50" or "gender = male". Let $c(X_n) \in \{0, 1\}$ be the indicator of whether data point $X_n$ satisfies the condition $c$.

#### 3.1.2 Rule.

A rule $r = (c_1, \ldots, c_L) \to z$ of *order* $L$ is a conjunction (logical AND) of $L$ conditions $c_1, \ldots, c_L$ conditions along with a predicted target outcome $z \in \{0, 1\}$. For example, the statement "If age $\geq$ 50 and BMI $\geq$ 30, then diabetes = 1" is a rule of order 2. Since this work seeks to explain when the black-box model predicts 1 with high confidence, this paper assumes that all rules $r$ predict $z = 1$; hence, we define a rule simply by $r = (c_1, \ldots, c_L)$.

A data point $X_n$ satisfies the rule $r$ if $c(X_n) = 1$ for all $c \in \{c_1, \ldots, c_L\}$ (i.e., the data point satisfies all conditions of the rule). Define $r(X_n)$ as the indicator of $X_n$ satisfying rule $r$.

$$r(X_n) = \begin{cases} 1 & \forall c_i \in r : c_i(X_n) = 1 \\ 0 & \text{otherwise} \end{cases} \tag{1}$$

#### 3.1.3 Rule List.

A rule list $\ell = (r_1, \ldots, r_M)$ of length $M$ is a disjunction (logical OR) of $M$ rules. $X_n$ satisfies the rule list $\ell$ if it satisfies as least one of the rules $r_1, \ldots, r_M$. Let $\ell(X_n)$ be the indicator of whether $X_n$ satisfies rule list $\ell$.

$$\ell(X_n) = \begin{cases} 1 & \exists r_i \in \ell : r_i(X_n) = 1 \\ 0 & \text{otherwise} \end{cases} \tag{2}$$

For notational convenience, let $[N]_\ell \subseteq [N]$ be the set of data points that satisfy $\ell$.

$$[N]_\ell := \{n \in [N] : \ell(X_n) = 1\} \tag{3}$$

Note that since the index $n \in [N]$ is also the ranking under the black-box model, $[N]_\ell$ contains the ranking of the data points that are satisfied by $\ell$. Since any rule $r$ can be considered a rule list of length 1, we also denote $[N]_r$ as the set of data points that satisfy $r$.

#### 3.1.4 Checklist.

A checklist $k_t = (c_1, \ldots, c_K)$ is defined by a list of $K$ conditions and a threshold $t \in \mathbb{N}$. A data point $X_n$ satisfies $k_t$ if it satisfies at least $t$ out of the $K$ conditions. Let $k_t(X_n)$ be an indicator for whether $X_n$ satisfies $k_t$

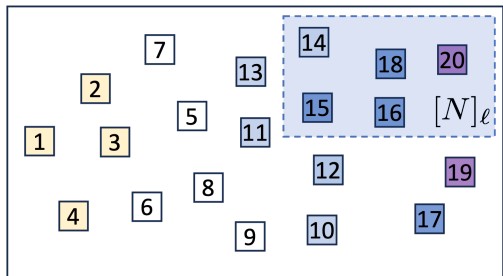

Figure 2: An example of $[N]_\ell$ for a rule list $\ell$ for $N = 20$, where the ranking of each point is marked. The support is $\texttt{supp}(\ell) = 5/20 = 0.25$ and the ABBR is $\texttt{abbr}(\ell) = \frac{1}{20} \cdot \text{avg}(\{14, 15, 16, 18, 20\}) = 0.83$

$$k_t(X_n) = \begin{cases} 1 & \sum_{i=1}^{K} c_i(X_n) \geq t \\ 0 & \text{otherwise} \end{cases} \tag{4}$$

A checklist is a special case of a rule list by listing out $\binom{K}{t}$ rules, one for each of the $\binom{K}{t}$ ways that $t$ conditions out of $K$ can be satisfied.

### 3.2 Evaluating a Rule List: Support, ABBR and Sparsity

A rule list $\ell$ (of which a checklist rule is a special case) provides a useful explanation of confident (i.e., highly ranked) black-box predictions if (1) $[N]_\ell$ is large in size, and (2) $[N]_\ell$ contains data points with large rank on average, and (3) $\ell$ is sparse (i.e., interpretable). We define each of the three criteria more formally.

#### 3.2.1 Support.

The support of $\ell$ is defined as the proportion of data points in $[N]_\ell$ (i.e., data points that satisfy $\ell$)

**Definition 1** (Support).

$$\boldsymbol{supp}(\ell) = \frac{1}{N} \cdot \big|[N]_\ell\big| \tag{5}$$

A rule list with higher support explains the model's behavior across more data points, and thus is more desirable.

#### 3.2.2 Average Black-Box Ranking

The ABBR of a rule list $\ell$ is simply the average ($\texttt{avg}$) value of the ranks $[N]_\ell$, normalized by $\frac{1}{N}$ to give a number between 0 and 1.

**Definition 2** (Average Black-Box Ranking).

$$\boldsymbol{abbr}(\ell) := \frac{1}{N} \cdot \boldsymbol{avg}([N]_\ell) \tag{6}$$

In addition to rule lists, Support and ABBR are defined on a rule by treating it as a rule list of length 1. An example calculation of support and ABBR is shown in Figure 2. Note that both high support and ABBR are needed for a useful rule– a rule that is satisfied by a single data point can have high ABBR, while a vacuous rule like "If (Age $\geq$ 0)" can easily be satisfied by the entire dataset.

#### 3.2.3 Sparsity.

For a rule list or checklist to be easily interpretable, sparsity is crucial. For a rule list, sparsity is measured in two ways: (1) the number of rules in the rule list, and (2) the maximum order (number of conditions) in each of those rules. For checklists, sparsity is measured by the number of conditions in the checklist. The algorithms developed in this paper will allow a user to explicitly set a limit on these measures of sparsity.

### 3.3 Connection between ABBR and Consistency.

There is a natural connection between ABBR, a novel metric calculated on the rankings of the model, and consistency, a traditional metric calculated on discrete $0/1$ predictions. Under $0/1$ predictions, a rule list $\ell$ that explains when a model predicts 1 over 0 can be evaluated based on its *consistency*, the proportion of $[N]_\ell$ that are predicted as 1. For any $m \in [N]$, the black-box's predictions can be discretized at rank $m$, such that $b(X_n) = 0$ if $n < m$ and $b(X_n) = 1$ for $n \geq m$. Equivalently, this is the proportion of $[N]_\ell$ with *rank at least $m$*.

**Definition 3** (Consistency with Threshold $m$)**.** *The consistency of $\ell$ to the predictions of $1$ with threshold chosen at rank $m$ is:*

$$consistency(\ell, m) := \frac{\left| [N]_\ell \cap \{m, m+1, \ldots, N\} \right|}{\left| [N]_\ell \right|} \tag{7}$$

Rather than subjectively choosing a particular threshold $m$ to discretize the black-box's predictions into 0 and 1, ABBR captures the *average consistency* across all possible thresholds. The following Lemma states this connection between ABBR and consistency.

**Lemma 3.1** (Equivalence to Average Consistency)**.** *For a rule list $\ell$, an equivalent definition of the Average Black-Box Ranking from Definition 2 is given by:*

$$abbr(\ell) := \frac{1}{N} \cdot \sum_{m=1}^{N} consistency(\ell, m) \tag{8}$$

A proof of this lemma can be found in Appendix B.

While ABBR is an intuitive metric generalizing the notion of consistency, we highlight the value of ABBR in Appendix A quantitatively by showing that rules which have higher ABBR tend to have better *generalizability* to unseen data, compared to rules that are just optimized for consistency with respect to an arbitrarily chosen threshold.

## 4 Optimizing for Rule Lists and Checklists

This section presents `RuleListSearch` and `ChecklistSearch`, algorithms that search for rule lists and checklists that maximize ABBR and support while satisfying user-defined criteria. The algorithms explicitly constrain the sparsity of the resulting rule list and checklists, as explained in Section 3.2.3. Section 4.1 formally defines this objective, while Section 4.2 presents an algorithmic framework of the two approaches. Details of the two algorithms are presented in Sections 4.3 and 4.4.

### 4.1 Objective: Maximize ABBR Given Support and Sparsity

This paper presents algorithms that take as input (1) sparsity constraints, and (2) a target support $s \in [0,1]$ and seek to find rule lists or checklists with high ABBR that satisfy both the sparsity constraints and has support of $s$. Using a specific support target $s$, it is possible to compare different algorithms; Given two rule lists or checklists with the same support, the one with the highest ABBR is preferred. However, a technical issue is that it is typically infeasible to find a rule list whose support is *exactly $s$* due to the discrete nature of a rule list. It is unlikely that there exists a rule list satisfied by exactly $s \cdot N$ data points (that is, not a single data point more or fewer). Furthermore, even if one could find such a rule with support $s$ on a particular dataset, its support on unseen data will certainly not be $s$, which complicates out-of-sample evaluation.

In light of this, let $abbr(\ell, s)$ be the ABBR of $\ell = (r_1, \ldots, r_M)$ interpolated at the target support $s$ for $s \neq supp(\ell)$. Let $\ell_m = (r_1, \ldots, r_m)$ be a sub rule list consisting of the first $m$ rules of $\ell$. Define this $m^*$ such that:

$$supp(\ell_{m^*}) \leq s < supp(\ell_{m^*+1}) \tag{9}$$

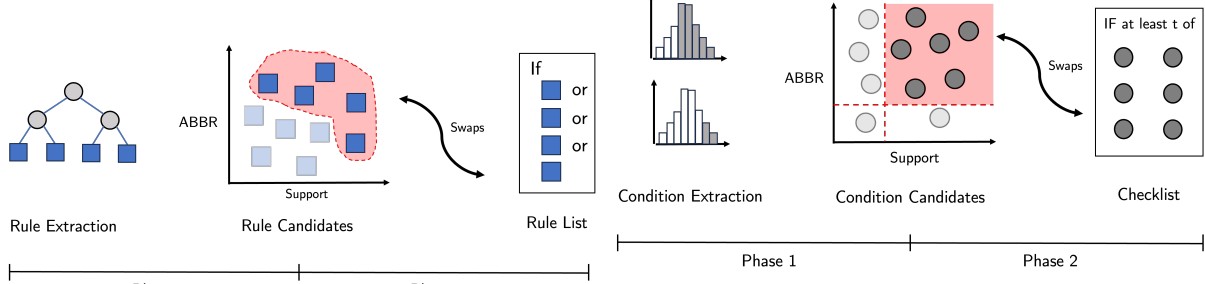

a **RuleListSearch**. A large set of rule $\overline{\mathcal{R}}$ is extracted using Random Forest (shown) or WizWhy (not shown), then filtered to $\mathcal{R}$ based on support and ABBR to generate a set of rule candidates. Local search is performed by randomly swapping a rules from the rule list with one from the set of rule candidates.

b **ChecklistSearch.** Conditions $\overline{\mathcal{C}}$ are extracted from each feature $d \in [D]$ by discretizing continuous features, then filtered to $\mathcal{C}$ based on support and ABBR to generate a set of condition candidates. Then, local search is performed by randomly swapping out a condition from the checklist and a condition from the set of condition candidates.

Figure 3: **Local Search Algorithmic Framework**.

This $m^*$ is chosen such that support of points satisfying the first $m^*$ rules is less than $s$, but adding those that satisfy $r_{m^*+1}$ causes the support to exceed $s$ [1].

Intuitively, $\mathtt{abbr}(\ell, s)$ is calculated by taking the average ranking of set with support exactly $s$, one that is induced by the ordering of the rules in $\ell$. This set is constructed by taking all the points in $[N]_{\ell_{m^*}}$ (whose support is $\mathtt{supp}(\ell_{m^*})$) and an arbitrary subset of the points in $[N]_{\ell_{m^*+1}} \setminus [N]_{\ell_m^*}$ such that this subset has support $s - \mathtt{supp}(\ell_{m^*})$. Across all choices of this arbitrary subset, the average ranking of this subset is equal to the average of the entire set $[N]_{\ell_{m^*+1}} \setminus [N]_{\ell_m^*}$. Let $\alpha := \frac{\mathtt{supp}(\ell_m^*)}{s}$ be the proportion of this reuslting set with support $s$ that came from $[N]_{\ell_{m^*}}$ instead of $[N]_{\ell_{m^*+1}} \setminus [N]_{\ell_m^*}$. The average ranking of this set of support exactly $s$ is given by the following convex combination.

$$\mathtt{abbr}(\ell, s) := \frac{1}{N} \left[ \alpha \cdot \mathtt{avg}([N]_{\ell_m^*}) + (1 - \alpha) \, \mathtt{avg}([N]_{\ell_{m^*+1}} \setminus [N]_{\ell_m^*}) \right]$$

## 4.2 Algorithmic Framework

The algorithms **RuleListSearch** and **ChecklistSearch** seek to find rule lists $\ell$ or checklist rule $k_t$ that maximize $\mathtt{abbr}(\cdot, s)$ as defined in Equation equation 4.1. Both **RuleListSearch** and **ChecklistSearch** follow the same 2 phase algorithmic framework based on local search.

1. Phase 1 generates a set of *rule candidates* $\mathcal{R}$ or *condition candidates* $\mathcal{C}$, which are the building blocks for selecting strong rule lists or checklists, respectively. This involves first generating a large set of rules $\overline{\mathcal{R}}$ or conditions $\overline{\mathcal{C}}$, then filtering them down to a manageable size by only keeping the rules or conditions that excel in support and ABBR.

2. Phase 2 performs a local search to maximize $\mathtt{abbr}(\cdot, s)$. Starting with an initial rule list or checklist, rules from the rule list or conditions from the checklist are randomly swapped out with those in the candidate set. Simulated annealing Kirkpatrick et al. (1983) is used to effectively balance exploration (that is, searching the space of possible rule lists or checklists) and exploitation (that is, making changes that improve the objective function).

Figure 3a and 3b provides a visual overview of Phase 1 and Phase 2 of **RuleListSearch** and **ChecklistSearch**. We dive into the details of the implementation below.

---

[1] Since these nested rule lists have increasing supports, $0 < \mathtt{supp}(\ell_1) < \mathtt{supp}(\ell_2) \leq \ldots \leq \mathtt{supp}(\ell_S) < 1$, this $m^*$ is well defined. As edge cases, we define $\ell_0$ as a rule satisfied by no data points (i.e., $\mathtt{supp}(\ell_0) = \mathtt{abbr}(\ell_0) = 0$) and add a dummy rule $r_{M+1}$ that is satisfied by all data points so that $\mathtt{supp}(\ell_{M+1}) = 1, \mathtt{abbr}(\ell_{S+1}) = 0.5$.

### 4.3 Details of `RuleListSearch`

`RuleListSearch` takes as parameters: (1) $s \in [0,1]$, the target support, (2) The maximum order of each rule $L$ and (3) The maximum length of the resulting rule list $M$.

Phase 1 of `RuleListSearch` first builds a large pool of rules $\overline{\mathcal{R}}$, which is then filtered to a refined set of candidate rules $\mathcal{R}$.

#### 4.3.1 Phase 1: Generating $\overline{\mathcal{R}}$.

We generate a large pool of rules $\overline{\mathcal{R}}$ in the following way.

**Case 1.** $L = 2$. Use WizWhy Meidan (2005), a software tool which exhaustively searches a dataset for rules of order up to $L$ and support at least $\gamma$. WizWhy relies on the Apriori Algorithm Agrawal et al. (1996), a well-known algorithm that was developed to efficiently frequent item-sets in market basket analysis (e.g., if a consumer buys bread and peanut butter, then they are likely to buy jelly). For higher order rules, exhaustive enumeration is typically computationally intractable.

**Case 2.** $L \geq 3$. Higher order rules are generated from leaves of Classification and Regression Trees (CART). That is, train a Random Forest Regressor on the dataset $\{X_n, \frac{n}{N}\}_{n=1}^N$, treating $\frac{n}{N}$ as a continuous dependent variable. The maximum depth of each tree in the random forest is set as $L$, the minimum proportion of data points in a leaf node is set as 0.02 and fix the number of trees at 50. Splits in each tree are chosen to minimize mean-squared error for predicting rank, which will separate data points with high and low rank. For each of the 50 trees, we extract a rule from each leaf node by noticing that any path from the root node to a leaf defines a list of conditions (e.g., Age $> 60$, BMI $< 35$). Choosing $L$ as the maximum depth of each tree ensures all rules extracted have order at most $L$, while choosing $\gamma$ as the minimum proportion of data points in a leaf ensures all rules have support at least $\gamma$.

#### 4.3.2 Phase 1: Filtering $\overline{\mathcal{R}}$ to $\mathcal{R}$.

The rules in $\overline{\mathcal{R}}$ are filtered to a more manageable size $\mathcal{R}$. Since $\overline{\mathcal{R}}$ was generated either exhaustively or via a heuristic, $\overline{\mathcal{R}}$ likely contains many rules with low ABBR. Intuitively, a rule list with high ABBR and support is composed of individual rules which also have high ABBR and support [2]. Since Phase 2 chooses rules from $\mathcal{R}$ to build rule lists, filtering $\overline{\mathcal{R}}$ into $\mathcal{R}$ improves the local search's chances of finding better rule lists.

The filtering criteria is defined below:

$$\mathcal{R} := \left\{ r \in \overline{\mathcal{R}} : \nexists r' \in \overline{\mathcal{R}} \text{ such that} \right.$$
$$\left. \mathrm{supp}(r') > \mathrm{supp}(r) \text{ and } \mathrm{abbr}(r') > 1.1 \cdot \mathrm{abbr}(r) \right\} \tag{10}$$

We add a rule $r \in \overline{\mathcal{R}}$ to $\mathcal{R}$ if it is *approximately Pareto optimal*, that is there does not exist another rule $r' \in \overline{\mathcal{R}}$ with strictly higher support and ABBR that is at least 1.1 times higher. A Pareto optimal rule is one that cannot be dominated in both support and ABBR, but our approximate Pareto optimal definition allows ABBR to be dominated by up to a factor of 1.1.

#### 4.3.3 Phase 2: Local Search

A local search procedure based on *simulated annealing* Kirkpatrick et al. (1983) is used to find a rule list $\ell$ with high $\mathrm{abbr}(\ell, s)$. Simulated annealing is a local search paradigm that effectively balances between exploration and exploitation. A starting rule list $\ell^{(0)}$ is constructed by taking the $M$ rules from $\overline{\mathcal{R}}$ with the highest ABBR. Over $T = 500$ iterations, in iteration $i \in [T]$, a candidate neighbor $\ell'$ is generated by swapping out a randomly chosen rule from $\ell^{(i-1)}$ and a rule from $\mathcal{R}$. To decide whether to update $\ell^{(i)}$ to $\ell'$ or keep $\ell^{(i)}$ at $\ell^{(i-1)}$, simulated annealing compares $\mathrm{abbr}(\ell', s)$ and $\mathrm{abbr}(\ell^{(i-1)}, s)$.

- If $\mathrm{abbr}(\ell', s) > \mathrm{abbr}(\ell^{(i-1)}, s)$ (i.e., $\ell'$ is an improvement over $\ell^{(i-1)}$), then update $\ell^{(i)}$ to be $\ell'$.

---

[2]Note that although we defined ABBR and support for rule lists in Section 3.2, the same metrics and be applied to an individual rule or condition by considering a rule as a rule list of length 1 and a condition as a rule of order 1.

- If $\mathtt{abbr}(\ell', s) \leq \mathtt{abbr}(\ell^{(i-1)}, s)$, then with some probability update $\ell^{(i)} = \ell'$, otherwise keep $\ell^{(i)} = \ell^{(i-1)}$. This probability of an update depends on (1) The difference in the objective function $\mathtt{abbr}(\ell^{(i-1)}, s) - \mathtt{abbr}(\ell', s)$ (i.e., how much worse $\ell'$ is compared to $\ell^{(i-1)}$) and (2) the value of $i$ relative to $T$ (i.e., how close the algorithm is to termination). For small values of $i$, simulated annealing is willing to explore and moves to neighbors $\ell'$ even if $\mathtt{abbr}(\ell', s) \leq \mathtt{abbr}(\ell^{(i-1),s}$, but as $i$ approaches $T$, the algorithm will exploit by only take steps that strictly increase $\mathtt{abbr}(\ell^{(i-1),s})$.

See Appendix D for details on the Simulated Annealing algorithm.

### 4.3.4 Final Rule List.

Let $\ell^{(T)} = (r_1, \ldots, r_M)$ be the output of the Simulated Annealing algorithm. Note that $\ell^{(T)}$ always has $M$ rules regardless of $s$, but some rules may be removed without affecting the ABBR. As a final step, $\mathtt{RuleListSearch}$ removes rules from the end of $\ell^{(T)}$ while keeping ABBR the same. Let $m^*$ be define as in equation equation 9, which states that $\mathtt{abbr}(s, \ell^{(t)})$ only depends on $\ell^{(T)}_{m^*}$ and $\ell^{(T)}_{m^*+1}$, we can remove all the rules $r_{m^*+2}, \ldots, r_M$ without changing the ABBR at support $s$. The final output of $\mathtt{RuleListSearch}$ is $\ell^{(T)}_{m^*+1}$.

### 4.4 Details of `ChecklistSearch`

`ChecklistSearch` takes as input (1) the target support $s \in [0, 1]$, and (2) the number of conditions $K$ in the checklist. `ChecklistSearch` follows the same 2-phase algorithmic framework as `RuleListSearch` except that it works with conditions instead of rules. Phase 1 builds a large pool of conditions $\overline{\mathcal{C}}$ and filters it down to $\mathcal{C}$, while Phase 2 performs local search for a checklist using $\mathcal{C}$ to generate neighbors.

### 4.4.1 Phase 1: Generate $\overline{\mathcal{C}}$ .

A pool of conditions is generated by going through each of the $D$ features. Recall that we assumed each feature is numerical (categorical variables have already been encoded).

- If $d$ is a discrete feature, add the conditions $(d, =, v)$ for all possible values $v$ in feature $d$ (e.g., Race = 1, Race = 2...etc.).

- If $d$ is a numerical feature, break the feature into 5 intervals based on the quantiles of the empirical distribution. Let $q_1, q_2, q_3, q_4$ be the $1/5$, $2/5$, $3/5$ and $4/5$ quantiles of the values $\{X_{nd}\}_{n=1}^{N}$. We add the conditions $(d, \geq, q_1), (d, \geq, q_1), \ldots, (d, \leq, q_2), (d, \geq, q_2)$ .

### 4.4.2 Phase 2: Filtering $\overline{\mathcal{C}}$ to $\mathcal{C}$.

For each condition, treat it as a rule with a single condition and evaluate its support and ABBR. We filter for conditions with a support at least 0.02 and no more than $1 - 0.02 = 0.98$ and ABBR of at least 0.5.

$$\mathcal{C} = \{c \in \overline{\mathcal{C}} : \mathtt{supp}(c) > 0.02, \mathtt{supp}(c) < 0.98, \mathtt{abbr}(c) \geq 0.5\} \tag{11}$$

Conditions satisfied by too few data points (e.g., Age > 95) or too many data points (e.g, Age < 95) are typically not useful in a checklist. Since `ChecklistSearch` seeks checklists with high ABBR, it removes conditions with ABBR less than 0.5 as those are correlated with predictions of 0 instead of 1.

### 4.4.3 Phase 3. Local Search

Perform a local search using simulated annealing identical to that from `RuleListSearch`. A starting checklist $k_t^{(0)}$ is chosen by to be the top $K$ conditions from $\mathcal{C}$ based on ABBR and computing a threshold $t$ to maximize the objective(i.e., $t = \arg\max_{t' \in [K]} \mathtt{abbr}(k_{t'}, s)$).

For each iteration $i \in [T]$, similar to the local search for rule lists, generate a neighbor $\tilde{k}_{t'}$ by randomly swapping one of the conditions in $k_t^{(i-1)}$ with one from $\mathcal{C}$ and choosing the appropriate threshold that maximizes ABBR

| Dataset Name | $N$ | $D$ | Example Features | Target Outcome | Black-Box AUC |
|---|---|---|---|---|---|
| Adults | 30,000 | 14 | Gender, age, ethnicity, education, marital status, occupation | Income >50K | 0.900 |
| Diabetes | 30,000 | 21 | BMI, age, smoker, high blood pressure, high cholesterol | Patient develops diabetes | 0.817 |
| FICO (Loan) | 10,459 | 23 | Risk estimate, month since last transaction, income | Default on the loan in next 2 years | 0.788 |
| Recidivism | 25,835 | 48 | gender, race, age at release, gang affiliated, education level | Commits a crime again in the next 3 years | 0.764 |
| Schizophrenia | 12,380 | 54 | Gene SPEN Pos 16262471 Gene NBPF1 Pos 16905788 | Patient is diagnosed with Schizophrenia | 0.675 |
| Readmission | 30,000 | 47 | hospital ID, admission type, length of stay, treatments received | Discharged patient was readmitted to the hospital within 30 days | 0.659 |

Table 1: **Overview of Datasets used in Empirical Experiments.** This tables shows the number of rows $N$, features $D$, examples of some features, the target outcome of interest and the AUC of the black-box Random Forest Classifier we seek to explain. Note that datasets with $N = 30,000$ have been sampled from a larger dataset to reduce computational time. Sources for the 6 datasets are as follows: Dua & Graff (2019a), Dua & Graff (2019b), FICO (2018), ProPublica (2016), for Biotechnology Information (2024), Dua & Graff (2019c)

at support $s$, that is $\tilde{t} = \arg\max_{t' \in [K]} \mathtt{abbr}(\tilde{k}_{t'}, s)$. Simulated annealing is used to determine whether to keep $k_t^{(i)} = k_t^{(i-1)}$ or update it to $k_t^{(i)} = \tilde{k}_{\tilde{t}}$.

## 5 Empirical Results

This section presents empirical results for `RuleListSearch` and `ChecklistSearch` on 6 real-world datasets for binary classification shown. All datasets are binary classification tasks chosen from a variety of application areas, including healthcare, finance, economics, and criminal justice. The 6 datasets are summarized in Table 1.

Our results are organized into two subsections:

- Section 5.1: Experiments that show our `RuleListSearch` and `ChecklistSearch` algorithms generate strong rules that achieve near-optimal ABBR on all 6 datasets. Ablation studies are performed to explain the value of higher order rules, longer rules, and interpretability.

- Section 5.2: Comparison of `RuleListSearch` to a state-of-the-art method from Rudin & Shaposhnik (2023), highlighting value of Phase 1 (candidate rule generation) of `RuleListSearch`.

### 5.1 Achieving Near-Optimal ABBR

**Setup.** Rule lists are generated with $L \in \{2, 3, 4\}$ (maximum order of each rule) and $M \in \{3, 5\}$ (maximum number of rules in the rule list). Checklists are generated with $K \in \{5, 7\}$ (number of conditions). The target support $s$ is chosen from $\{0.1, 0.2\}$. The black-box predictions $\{b(X_n)\}_{n=1}^N$ are generated from a Random Forest Classifier trained with 500 estimators. All empirical results are reported out-of-sample on a holdout set of data not used in finding the rule list or checklist. Experiments were run on MIT's Engage Compute Cluster over 100 train/test splits, taking around 1 hour of compute time when parallelized with 24 cores. Each instance of `RuleListSearch` or `ChecklistSearch` take no more than 10 seconds to run on the 6 datasets in Table 1. Results shown are the average of the test performance across the 100 train/test splits.

**Target Support = 10%**

| Dataset | | L=2 | L=3 | L=4 | Best ★ | K=5 | K=7 |
|---|---|---|---|---|---|---|---|
| | | Rule Lists | | | | Checklists | |
| Recidivism | M=3 | -6.6% | -3.5% | 0.1% | 92.4% | 90.4% | 91.0% |
| | M=5 | -6.5% | -3.9% | ★ | | | |
| FICO | M=3 | -9.1% | -3.5% | -0.4% | 92.8% | 91.7% | 92.7% |
| | M=5 | -9.7% | -3.3% | ★ | | | |
| Adults | M=3 | -2.2% | -0.2% | -0.1% | 98.2% | 94.0% | 95.0% |
| | M=5 | -2.6% | -0.2% | ★ | | | |
| Diabetes | M=3 | -9.8% | -3.7% | -0.3% | 96.6% | 95.6% | 95.6% |
| | M=5 | -9.7% | -3.4% | ★ | | | |
| Schizo | M=3 | -4.7% | -1.3% | -2.4% | 97.2% | 96.5% | 98.9% |
| | M=5 | -4.9% | -0.3% | ★ | | | |
| Readmission | M=3 | -9.4% | -4.4% | -0.4% | 86.3% | 81.0% | 85.5% |
| | M=5 | -8.8% | -4.0% | ★ | | | |

**Target Support = 20%**

| Dataset | | L=2 | L=3 | L=4 | Best ★ | K=5 | K=7 |
|---|---|---|---|---|---|---|---|
| | | Rule Lists | | | | Checklists | |
| Recidivism | M=3 | -6.2% | -4.9% | -3.5% | 91.6% | 85.4% | 89.8% |
| | M=5 | -6.5% | -2.8% | ★ | | | |
| FICO | M=3 | -8.3% | -2.9% | -1.8% | 94.1% | 92.3% | 92.4% |
| | M=5 | -7.6% | -2.0% | ★ | | | |
| Adults | M=3 | -2.9% | -1.5% | -1.4% | 95.9% | 90.8% | 93.5% |
| | M=5 | -0.4% | 0.0% | ★ | | | |
| Diabetes | M=3 | -8.2% | -0.5% | -0.2% | 94.8% | 94.1% | 97.0% |
| | M=5 | -7.1% | -0.3% | ★ | | | |
| Schizo | M=3 | -4.2% | -2.1% | -1.7% | 84.7% | 88.8% | 89.2% |
| | M=5 | -2.0% | -0.2% | ★ | | | |
| Readmission | M=3 | -9.1% | -2.8% | -0.2% | 87.0% | 86.0% | 87.0% |
| | M=5 | -9.7% | -1.9% | ★ | | | |

Table 2: Optimality ratio of various methods for $s = 0.10$ (left) and $s = 0.20$ (right). In each table, rule lists are generated at $M \in \{3, 5\}$ and $L \in \{2, 3, 4\}$, with the best optimality ratio (which is always $L = 4, S = 5$) shown in the "Best" column. Shaded cells represent the absolute difference in optimality ratio from the "Best" column. For checklists, the optimality ratio is shown for $K = 5$ and $K = 7$. The best algorithm in each row is shown in blue text.

For each generated rule list or checklist, we report its *optimality ratio*, defined to be the ratio between its improvement over a naive baseline versus the improvement of an *optimal* rule relative to a naive baseline. A rule list with support $s$ can have a maximum possible ABBR of $1 - \frac{1}{2} \cdot s$, which is achieved if the rule list is satisfied by *exactly* the top $s$ percentile of the dataset (i.e., those with ranking $(1 - s) \cdot N$ through $N$, for an ABBR of $1 - \frac{1}{2} \cdot s$). A naive baseline for ABBR is 0.5 as a randomly chosen subset of the data will have ABBR on average 0.5. Hence, the following definition:

$$\text{Optimality Ratio}(\ell, s) := \frac{\texttt{abbr}(\ell, s) - 0.5}{1 - \frac{1}{2} \cdot s - 0.5} \tag{12}$$

Table 2 presents the main results of this empirical study, with $s = 0.10$ shown in the left table and $s = 0.20$ shown in the right table. Results for `RuleListSearch` are reported under "Rule Lists". which varies $L \in \{2, 3, 4\}$ and $M \in \{3, 5\}$ in a matrix. The value of $L$ and $M$ that generates the top optimality ratio is marked with a star and its optimality ratio is reported in the column "Best", while the colored cells represent the absolute delta with the optimality ratio of "Best". For `ChecklistSearch`, the two columns under "Checklists" show the optimality ratio for $K = 5$ and $K = 7$. In both tables, the algorithm with the highest optimality ratio is highlighted in blue.

It is clear that `RuleListSearch` and `ChecklistSearch` exhibit strong performance across 6 datasets. Across $s = 0.10$ and $s = 20$, the best rule list achieves an *average* optimality ratio of 0.926 (an optimality gap of 7.4%), while the best checklist achieves an average optimality ratio of 0.923 (an optimality gap of 7.7%). Some examples of the resulting rule lists and checklists for each dataset are shown in Figures 6 and 7 in Appendix F.

### 5.1.1 Sparsity vs. ABBR Trade-off in Rule Lists

Although larger values of $L$ and $M$ result in less sparsity and lower interpretabilty in a rule list, they give rise to rules with higher ABBR for any support. This trade-off between sparsity and performance is evident

| $L$ | Rule | Support | ABBR |
|---|---|---|---|
| 2 | If Heart Disease = 1 and High Cholesterol = 1 | 0.07 | 0.902 |
| 3 | If Difficulty Walking = 1, High Cholesterol = 1 and BMI > 29 | 0.05 | 0.943 |
| 4 | If BMI > 29, General Health < 3, High Blood Pressure = 1 and Heart Disease = 1 | 0.03 | 0.964 |

Table 3: Rules in $\mathcal{R}$ with the Highest ABBR for $L \in \{2, 3, 4\}$ (Diabetes Dataset)

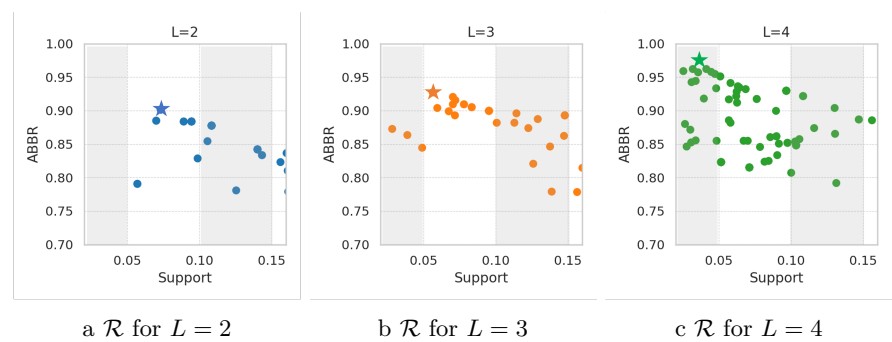

a $\mathcal{R}$ for $L = 2$      b $\mathcal{R}$ for $L = 3$      c $\mathcal{R}$ for $L = 4$

Figure 4: **Support and ABBR of Rules in $\mathcal{R}$ for various $L$. in the FICO Dataset** Larger value of $L$ lead to rules in $\mathcal{R}$ with higher ABBR and thus leader to higher ABBR when combined into rule lists.

.

in Table 2 as the optimality ratio is generally increasing in both $L$ and $M$, with the best optimality ratio achieved at $L = 4$ and $M = 5$.

**Value of Higher Order Rules.** First, we study the impact of rules of higher order (increasing $L$). Higher order rules are able to identify more specific subsets of the data, including ones where all data points are ranked more highly by the black-box model. This can be seen by examining the pool of rules $\mathcal{R}$ that `RuleListSearch` uses to construct rule lists. Figure 4a, 4b and 4c plot the support ($x$-axis) and ABBR ($y$-axis) of the rules in $\mathcal{R}$ for $L = 2, 3$ and 4 in the diabetes dataset. As $L$ increases, more rules appear in the *high ABBR but lower support region* (i.e., top left region in the figures). Table 3 shows the rule with the *highest* ABBR in each of the 3 rule candidate sets, which is marked with a star. While the rule for $L = 2$ is fairly broad (i.e., there are many patients with both heart disease and high cholesterol), the rule for $L = 4$ is much more specific and thus has a lower support, but captures a subset of the population which on average are at very high predicted risk of diabetes. Intuitively, it is unlikely that a rule with just 2 conditions can identify a set of patients, all of which are at high risk of diabetes. Black box model typically predicts confidently on a data point due to the existence of numerous risk factors, and hence explaining such data points requires higher order rules that can filter for patients with multiple risk factors. Since $\mathcal{R}$ for $L = 4$ has more rules with high ABBR than $L = 2, 3$, it is clear that by combining such rules together, we can get rule lists with better ABBR.

**Value of Longer Rule Lists $M$.** Given that rules with higher ABBR tend to have lower support, the length of a rule list $M$ is also crucial. As $M$ increases, the rule list can fit more rules with low support and high ABBR, whereas small values of $M$ force each rule of the rule list to have higher support and in turn a lower ABBR. This is evident in Table 2 as the benefit of $M = 5$ over $M = 3$ is more prominent for $s = 0.20$ over $s = 0.10$. To achieve a target support of $s = 0.20$ with $M = 3$ rules, each rule must have on average a support of 0.066, thus compromising its ABBR. On the other hand, when $M = 5$, each rule must only have a 4% coverage on average, allowing us to choose rules with lower support and higher ABBR. Similarly, the optimality ratios of the best rule lists are higher for $M = 5$ when $s = 0.10$ over $s = 0.20$ for the same reasoning. Given a fixed $M$, a lower target support allows the rule list to use low-support, but high ABBR rules, while a larger target support precludes a rule list from using such rules.

### 5.1.2 Improving Interpretability with Checklists

Although rule lists perform well, rule lists with $M = 5$ rules and $L = 4$ conditions per rule may be difficult to interpret. Checklists are designed with greater interpretability in mind. Table 2 shows that while checklists tend to slightly under-perform rule lists at $s = 0.10$, they can sometimes match or exceed rule lists at $s = 0.20$. This can be explained by discussing the tradeoffs between rule lists and checklists.

**Advantage of Checklists.** The main advantage of checklists over rule lists is its compact form and interpretability. A rule list with $M = 5$ and $L = 4$ can have 5 rules with 4 conditions each, resulting in a verbose list involving up to 20 conditions. On the other hand, the compact representation of a checklist allows it to implicitly define rule lists involving many rules without being difficult to interpret. For example, a checklist with $K = 7$ and $T = 5$ implicitly defines a rule list with $\binom{7}{5} = 14$ rules. While the rule lists in our experiment are constrained to a maximum of $M = 5$ rules, a checklist can bypass this constraint as it can implicitly define rule lists with many more rules. This is especially advantageous when the target support is high, hence why checklists perform better at $s = 0.20$ than $s = 0.10$.

**Disadvantage of Checklists.** The main disadvantage of checklists is that by assuming an equally weighted linear relationship between each of its $K = 5$ conditions, the checklist may inadvertently capture combination of conditions that have low ABBR. For example, consider a checklist $\ell_3 = (c_1, c_2, c_3, c_4)$, a checklist with $K = 4$ conditions and threshold $t = 3$. This checklist is equivalent to a rule list with the following 4 rules $r_1 = (c_1, c_2, c_3)$, $r_2 = (c_1, c_2, c_4)$, $r_3 = (c_1, c_3, c_4)$ and $r_4 = (c_2, c_3, c_4)$. It may be the case that while $\texttt{abbr}(r_1), \texttt{abbr}(r_2)$ and $\texttt{abbr}(r_3)$ are high, $\texttt{abbr}(r_4)$ is relatively low. Intuitively, this means that $c_1$ is a very crucial condition that should be weighted more than the other conditions. While a rule list would be able to explicitly list out rules $r_1, r_2$ and $r_3$, a checklist is forced to also include $r_4$, thereby dragging down the overall ABBR.

At low target support (i.e., $s = 0.10$), the disadvantage of checklists outweigh the advantages as rule lists can better capture a specific list of rules with high ABBR. On the other hand, higher target supports like $s = 0.20$ start to favor checklists for some datasets as longer rule lists are needed for rule lists to achieve this support, but rule lists are constrained by $M = 5$.

## 5.2 Comparison to Baseline Method

In this section, we propose a natural baseline algorithm to compare against `RuleListSearch`. First, recall that the `RuleListSearch` algorithm consists of two broad steps:

1. Mining for a set of candidate rules through Random Forest. Candidate rules are chosen to optimize ABBR while satisfying a minimum support.

2. Combining the rules together to form a Rule List, using simulated annealing to optimize for the in-sample ABBR at a target support.

Note that in the body of the paper, we refer to Step 1 above as Phase 1 (mining) and Phase 2 (filtering rules), while we referred to Step 2 as Phase 3 (optimization).

### 5.2.1 Benchmark Description

We compare `RuleListSearch` with a benchmark where we replace Step 1. above with a state-of-the-art method of generating a set of rules, in particular Rudin & Shaposhnik (2023)

Rudin & Shaposhnik (2023)'s work *Globally-Consistent Rule-Based Summary-Explanations for Machine Learning Models* constructs local explanations for a given data point by finding a rule that is globally consistent, i.e. matches the predictions of a black-box algorithm entirely. For example, if a patient where a black-box model predicts the patient will develop diabetes, a globally consistent explanation for this patient is an if-then rule under which *all* patients are predicted by the model to develop diabetes.

While Rudin & Shaposhnik (2023) generates local explanations for a given data instance, we can aggregate these rules into a global explanation using Step 2. Our benchmark approach is to replace Step 1 above with a

rule-mining procedure where we take the predictions of a black-box algorithm, discretize them at $p \in [0, 1]$, then construct a globally consistent local explanation for every data point for which the black-box model's predicted probability is greater than $p$. We feed this large set of rules into Step 2, which outputs a rule list while optimizing for ABBR. This is unchanged from still `RuleListSearch`.

### 5.2.2 Results and Discussion

We evaluate this baseline using two discretization thresholds: $p = 0.5$ (a natural threshold) and $p = 0.9$ (an aggressive threshold). We set a minimum support of 0.1 for any rule in the pool of rules. Table 4 reports confidence intervals for the ABBR of the final rule list after simulated annealing, comparing the baseline against `RuleListSearch`. Note that we were unable to generate results the benchmark in the $p = 0.9$ case under a reasonable time limit, due to a limitation discussed below.

| Dataset Name | Our Algorithm | Benchmark ($p = 0.5$) | Benchmark ($p = 0.9$) |
|---|---|---|---|
| Adults | $0.941 \pm 0.031$ | $0.697 \pm 0.062$ | N/A |
| Diabetes | $0.933 \pm 0.010$ | $0.912 \pm 0.007$ | N/A |
| FICO | $0.913 \pm 0.020$ | $0.862 \pm 0.018$ | N/A |
| Readmission | $0.880 \pm 0.003$ | $0.861 \pm 0.008$ | N/A |
| Recidivism | $0.910 \pm 0.008$ | $0.886 \pm 0.012$ | N/A |
| Schizo | $0.935 \pm 0.014$ | $0.903 \pm 0.011$ | N/A |

Table 4: Performance comparison between our algorithm and benchmark models with different $p$ values.

The results reveal fundamental limitations of approaches that rely solely on binary prediction alignment:

- **Natural Threshold ($p = 0.5$)**: When using a moderate cutoff, Rudin & Shaposhnik (2023)'s method successfully generates globally consistent rules, but these rules achieve significantly lower ABBR scores compared to `RuleListSearch`. This occurs because global consistency with binary predictions fails to distinguish between instances with vastly different model confidence levels. For instance, a patient with 51% predicted probability of diabetes is treated identically to a patient with 99% predicted probability, despite the substantial difference in model certainty.

- **Aggressive Threshold ($p = 0.9$)**: With a high threshold, Rudin & Shaposhnik (2023)'s approach fails to find globally consistent rules altogether. The requirement for 100% consistency becomes nearly impossible to satisfy when targeting only very high-confidence predictions, as real-world datasets rarely contain perfect decision boundaries that achieve complete separation. Rudin & Shaposhnik (2023) leverages an integer optimization program to find globally consistent rules involving up to $k$ features, which becomes intractable when $k$ is large.

This benchmark experiment demonstrates that existing approaches focusing on alignment with 0/1 predictions ignore the confidence of those predictions, leading to either (1) rules that don't explain confidence, or (2) the lack of any rules at all. `RuleListSearch` fixes both of these by focusing on rules that explain high-confidence predictions and mining for such rules using a heuristic (i.e., Random Forest) rather than an optimization approach.

## 6 Conclusion

This paper presented a novel evaluation metric for model-agnostic post-hoc explanations that emphasizes when a model predicts confidently. Rather than evaluate an explanation on how consistent it is with the binary predictions of black-box model, the ABBR metric rewards rules that capture a large number of *confident* predictions. This paper argues that explaining confident predictions is more useful than just explaining when a model a prediction, which can possibly be a guess. With this goal in mind, this paper developed methodologies for finding rule lists and checklists, both of which were based on a local search heuristic. Empirical results show that rule lists with up to 5 rules achieve strong performance, attaining an ABBR

within 7.4% of the optimal possible ABBR on average, while checklists can achieve better interpretability with a slight reduction in performance.

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

## A    Justification of ABBR as a Useful Metric

In this section, we justify the usefulness of ABBR as a metric of explainability by arguing that *rules with higher ABBR tend to be more generalizable to unseen data.* Generalizability is crucial because explanations that overfit to specific training examples may give incorrect insights about the model's true decision boundary, leading to incorrect conclusions downstream. We leverage *consistency* with the 0/1 predictions of a black-box model as a well-established baseline metric. We show that rules optimized for in-sample ABBR exhibit better generalization than rules optimized for consistency alone.

### A.1    Experiment Setup

We define the *consistency generalization gap* of rule $r$ as:

$$\text{Gap}(r) = \text{Consistency}(r, y_{train}) - \text{Consistency}(r, y_{test})$$

where $y_{train}$ and $y_{test}$ are binary predictions from a black-box model with some threshold $p$. This gap is typically positive, as rules are selected based on their in-sample performance. Recall that the consistency of a rule $r$ with respect to binary predictions is simply the fraction of the data points that $r$ applies to for which $y = 1$.

Our exact experimental procedure is as follows:

1. Generate 1000 random rules of up to 5 conditions by randomly sampling 5 features, and choosing thresholds (continuous) or values (discrete) for those features. Only keep rules with a minimum support of at least 2%.

2. Out of these 1000 rules, let $r_a$ be the rule with the highest in-sample ABBR, and let $r_c$ be the rule with the highest in-sample consistency.

3. We report the following: the precent decrease in the gap of $r_{abbr}$ relative to $r_{consistency}$.

$$\frac{\text{Gap}(r_{consistency}) - \text{Gap}(r_{abbr})}{\text{Gap}(r_{consistency})} \tag{13}$$

### A.2    Results and Discussion

| Dataset Name | Gap($r_{abbr}$) | Gap($r_{consistency}$) | % Improvement |
|---|---|---|---|
| Adults | $0.3\% \pm 0.1\%$ | $1.1\% \pm 0.2\%$ | 72.7% |
| Diabetes | $0.1\% \pm 0.0\%$ | $0.5\% \pm 0.0\%$ | 80.0% |
| FICO | $1.0\% \pm 0.3\%$ | $2.8\% \pm 1.1\%$ | 64.3% |
| Readmission | $0.7\% \pm 0.1\%$ | $1.0\% \pm 0.1\%$ | 30.0% |
| Recidivism | $0.2\% \pm 0.3\%$ | $1.1\% \pm 0.3\%$ | 81.8% |
| Schizo | $1.5\% \pm 0.2\%$ | $1.0\% \pm 0.2\%$ | $-50.0\%$ |

Table 5: Generalization gap comparison between ABBR-Rules and Consistency Rules

Table 5 shows the 95% confidence interval of $\text{Gap}(r_{consistency})$ and $\text{Gap}(r_{abbr})$ across 500 test/trains splits. The last column shows the percent improvement of the ABBR-rules versus the consistency rules. Across the 6 datasets, 5 of them show better generalizability in the ABBR optimized rules compared to the consistency optimized rules.

The key insight behind ABBR's superior generalizability lies in how binary classification thresholds create artificial boundaries. When discretizing predicted probabilities using a threshold $p$, instances near the decision boundary are arbitrarily classified. For instance, with $p = 0.9$, an instance with probability 0.89 receives label 0 while probability 0.91 receives label 1, despite minimal difference in model confidence. Rules that rely heavily on such boundary cases are prone to poor generalization. ABBR mitigates this by considering the underlying probabilities rather than just binary agreement.

# B Proof of Lemma 3.1

*Proof.* Proof of Lemma 3.1 We leverage a well-known fact: For any random variable $Z$ which takes on non-negative values with cumulative distribution function $F_Z(z) := P(Z < z)$, one way of calculating the expectation of $Z$ is given by

$$E[Z] = \int_0^\infty (1 - F_Z(z))dz \tag{14}$$

Take $Z$ to be a discrete random variable that takes on a value from the set $[N]_\ell$ uniformly at random. Then, $E[Z]$ is simply the ABBR metric, whereas $1 - f_Z(z)$ is simply $\texttt{consistency}(\ell, z)$. Finally, noting that the integral from 0 to $\infty$ can be replaced with a summation due to the step-function nature of $f_Z(z)$, we obtain the desired result. $\square$

# C Pseudo-Code for `RuleListMiner` and `ChecklistMiner`

Pseudo-code for `RuleListSearch` and `ChecklistSearch` are provided in Algorithm 1 and Algorithm 2 below. The function `Simulated Annealing` is defined in Equation equation 15 in Appendix D.

---

**Algorithm 1** `RuleListSearch`

---

1: **Input:** $\{X_{nd}\}_{n \in [N], d \in [D]}, s \in [0, 1], M \in \mathbb{N}$(max number of rules), $L \in \mathbb{N}$(maximum order of each rule)
2: **Output:** Rule List $\ell = (r_1, \ldots, r_m)$ with $m \leq M$ and $|r_i| \leq L$.
3:
4: **if** $L \geq 3$ **then**
5:     $\overline{\mathcal{R}} \leftarrow \{\}$                                                          ▷ Large Pool of Rules
6:     **for** $\texttt{tree} \in \texttt{RandomForestRegressor} \left( \left\{ X_n, \frac{n}{N} \right\}_{n=1}^N, L, 0.02 \right)$ **do**
7:         **for** $\texttt{leaf} \in \texttt{tree}$ **do**
8:             $\overline{\mathcal{R}} \leftarrow \mathcal{R} \cup \text{Rule}(\texttt{leaf})$
9:         **end for**
10:     **end for**
11: **else**
12:     $\overline{\mathcal{R}} \leftarrow \text{WizWhy}(\{X_n\}_{n=1}^N, 0.02)$
13: **end if**
14:
15: $\mathcal{R} = \{\}$                                                              ▷ Filtered Pool of Rules
16: **for** $r \in \overline{\mathcal{R}}$ **do**
17:     **if** $\forall r' \in \overline{\mathcal{R}}$ it is true that $\texttt{supp}(r') \leq \texttt{supp}(r)$ or $\texttt{abbr}(r') \leq 1.1 \cdot \texttt{abbr}(r)$ **then**
18:         $\mathcal{R} \leftarrow \mathcal{R} \cup r$
19:     **end if**
20: **end for**
21: $\mathcal{R} \leftarrow \text{sort}(\mathcal{R}, \texttt{abbr})$                                       ▷ Descending order of ABBR
22:
23: $\ell^{(0)} = \mathcal{R}[: N]$                                      ▷ Take first $S$ rules to form a rule list
24: **for** $i \in [T]$ **do**
25:     $n_1 \leftarrow \text{rand}(1, 2, \ldots, S)$
26:     $n_2 \leftarrow \text{rand}(1, 2, \ldots, |\mathcal{R}|)$
27:     $\ell' \leftarrow \ell_{(i)}$                                           ▷ Candidate Neighbor to Move To
28:     $\ell'[n_1] \leftarrow \mathcal{R}[n_2]$
29:     $\ell_{(i)} \leftarrow \text{Simulated Annealing}(\texttt{abbr}(\ell_{(i-1)}, s), \texttt{abbr}(\ell', s), i, T)$
30: **end for**
31: **Output.** $\ell_{m*+1}^{(T)}$, where $m^*$ is chosen such that $\texttt{supp}(\ell_{m^*}) \leq s < \texttt{supp}(\ell_{m^*+1})$.

---

---

**Algorithm 2** `ChecklistSearch`

---

1: **Input:** $\{X_{nd}\}_{n\in[N],d\in[D]}, s \in [0,1]$
2: **Parameters:** $K \in \mathbb{N}$ (number of conditions)
3: **Output:** Checklist $k_t = (c_1, \ldots, c_K)$.
4:
5: $\overline{\mathcal{C}} \leftarrow \{\}$                                                                       ▷ Large Pool of Conditions
6: **for** $d \in [D]$ **do**
7:     $\overline{\mathcal{C}} \leftarrow \overline{\mathcal{C}} \cup \text{Conditions}(d, Q)$                                                ▷ Discretize or One-Hot-Encode
8: **end for**
9:
10: $\mathcal{C} = \{\}$                                                                                      ▷ Filtered Pool of Conditions
11: **for** $c \in \overline{\mathcal{C}}$ **do**
12:     **if** $\text{supp}(c) \geq 0.02$ & $\text{supp}(c) \leq 0.98$ & $\text{abbr}(c) \geq 0.5$ **then**
13:         $\mathcal{C} \leftarrow \mathcal{C} \cup c$
14:     **end if**
15: **end for**
16: $\mathcal{C} \leftarrow \text{sort}(\mathcal{C}, \text{abbr})$                                                              ▷ Descending order of ABBR
17:
18: $k_t = \mathcal{C}[: N], t = \arg\max_{t'\in[K]} \text{abbr}(k_t, s)$                          ▷ Take first $K$ conditions to form checklist
19: **for** $i \in [500]$ **do**
20:     $n_1 \leftarrow \text{rand}(1, 2, \ldots, K)$
21:     $n_2 \leftarrow \text{rand}(1, 2, \ldots, |\mathcal{C}|)$
22:     $\tilde{k} \leftarrow k$                                                                           ▷ Candidate Neighbor to Move To
23:     $\tilde{k}[n_1] \leftarrow \mathcal{R}[n_2]$
24:     $\tilde{t} \leftarrow \arg\max_{t'\in[K]} \text{abbr}(k'_{t'}, s)$
25:     $k_t^{(i)} \leftarrow \text{SA}(\text{abbr}(k_t^{(i-1)}, s), \text{abbr}(\tilde{k}_{\tilde{t}}, s), i, T)$
26: **end for**
27: **Output.** $k_t^{(T)}$

---

$$x_{(i)} = \text{Simulated Annealing}(f(x_{(i-1)}), f(x'), i, T) := \begin{cases} x' & U < \exp\left\{K \cdot \frac{i}{T} \cdot [f(x') - f(x_{(i-1)})]\right\} \text{ where } U \sim \text{Unif}(0,1) \\ x_{(i-1)} & \text{otherwise} \end{cases}$$

(15)

Figure 5: Simulated Annealing Updating

## D   Details of Simulated Annealing

Simulated annealing is a probabilistic local search heuristic which optimizes an objective function through finding candidate neighbors and moving to them with probability. It is often used for optimizations with a large and discrete search space. Suppose we are trying to maximize a function $f(x)$ where $x \in \mathcal{X}$ is in input in some high-dimensional space. Simulated annealing works by first starting with solution $x_{(0)}$ and slowly updating it over the course of $T$ iterations. In iteration $i \in [T]$, the algorithm considers $x_{(i-1)}$ and generates a *neighbor* $x'$ by typically applying a small perturbation to $x_{(i-1)}$. We compare $f(x_{(i-1)})$ with $f(x')$ and decide to either (1) update $x_{(i)}$ to $x'$ or (2) keep $x_{(i)}$ equal to $x_{(i-1)}$. This decision is made probabilistically as shown in Figure 5.

Here, we draw a value $U \sim \text{Unif}(0,1)$ and compare it to the quantity $\exp\left(-K \cdot \frac{i}{T} \cdot [f(x') - f(x_{(i-1)})]\right)$, where $K$ is a hyper-parameter. This quantity can be dissected as follows:

- If $f(x') - f(x_{(i-1)}) > 0$ (i.e., $x'$ is an improvement over $x_{(i-1)}$), then $\exp\left\{K \cdot \frac{i}{T} \cdot [f(x') - f(x_{(i-1)})]\right\} > 1$ and so we will always choose $x_{(i)} = x'$.

- Otherwise $f(x') - f(x_{(i-1)}) < 0$, then we will choose $x_{(i)} = x'$ with probability that is dependent on: (1) How negative $f(x') - f(x_{(i-1)})$ is and (2) The quantity $\frac{i}{T}$. Intuitively, if $f(x')$ is significantly worse than $f(x_{(i-1)})$, then we will choose $x'$ with small probability. Also, as the iteration number $i$ increases, we will chooses also choose to move to suboptimal candidates $x'$ less frequently.

In our paper, we fix $K = 0.03$.

## E   Datasets Descriptions

We provide an overview of the datasets used in our empirical study.

- **Recidivism** ProPublica (2016): Data collected using the Correctional Offender Management Profiling for Alternative Sanctions (COMPAS), an algorithm used in aiding a judge or parole officer's decision in granting/denying parole to a defendant.

- **FICO (Loan Approval)** FICO (2018): As part of an interpretable machine learning challenge, this dataset consists of Home Equity Line of Credit (HELOC) applications made by homeowners and whether they repaid the loan within 2 years. Such a model is used to make real-time decisions on whether to approve or reject loan applications.

- **Diabetes** Dua & Graff (2019b): Compiled by the Center for Disease Control (CDC) in 2014, the Diabetes Health Indicators dataset is used to find lifestyle and demographic information that correlate with getting diabetes.

- **Adults Income** Dua & Graff (2019a). This is a classic dataset compiled by Barry Becker from the 1994 Census database. It is used for a variety of machine learning benchmarking research. Each row correspond to a particular individual and the goal is to predict whether that individual's annual income is greater than $50,000.

- **Schizophrenia** for Biotechnology Information (2024). The Sweden-Schizophrenia Population-Based Case-Control Exome Sequencing dataset, available through Database for Genotypes and Phenotypes

If any of the following rules are satisfied:

  Prior Arrests $\geq$ 3.5, Drug Tests Positive $\geq$ 0.06, Jobs Per Year $\leq$ 0.23, Age at Release $\leq$ 48

  Employed $\leq$ 0.47, Prior Arrests $\geq$ 2, Gang Affiliated = 1, Prior Misdemeanors $\geq$ 4

  Gang Affiliated = 1, Prior Arrests $\geq$ 1, Drug Tests $\geq$ 0.04, Days Employed $\leq$ 0.47

then predict 1 to recidivism

a Recidivism: Support = 0.104, ABBR = 0.913

If any of the following rules are satisfied:

  External Risk Estimate $\leq$ 65.5, Average Months in File $\leq$ 58.5, Trades with Balance $\geq$ 80.5

  External Risk Estimate $\leq$ 70.5, Average Months in File $\leq$ 59.5, Trades with Balance $\geq$ 79.5 and Months Since Oldest T

  Percent Trades Never Delinquent $\leq$ 91.5, External Risk Estimate $\leq$ 66.5, Average Moths in File $\leq$ 59.5

then predict 1 to FICO loan default.

b FICO: Support = 0.102, ABBR = 0.910

If any of the following rules are satisfied:

  Relationship = Husband, Education $\geq$ 14, Age $\geq$ 30

  Married = 1, Capital Gain $\geq$ 5000

  Sex = Male, Married = 1, Work Hours Per Week $\geq$ 42, Education $\geq$ 13

then predict 1 to Income $\geq 50,000$.

c Adults: Support = 0.105, ABBR = 0.941

If any of the following rules are satisfied:

  High Blood Pressure = 1, High Cholesterol = 1, BMI $\geq$ 29.5, General Health $\geq$ 3

  High Blood Pressure = 1, BMI $\geq$ 27.5, High Cholesterol = 1, Difficulty Walking = 1

  High Blood Pressure, General Health $\geq$ 27.5, BMI $\geq$ 28, Heart Disease = 1

then predict 1 to diabetes

d Diabetes: Support = 0.110, ABBR = 0.927

If any of the following rules are satisfied:

  Admission Type $\neq$ 3, Insulin = Yes and Diagnosis = 428

  Number Inpatient = 0, Admission Type = 6 and Diabetes Meds = 1

  Insulin = Yes, Admission Type = 6 and Number Inpatient = 1

then predict 1 to readmission

e Readmission: Support = 0.08, ABBR = 0.83

Figure 6: Examples of Rule Lists for $M = 3, L = 4$

(DbGaP) from the National Institute of Health (NIH) contains mutation-level genomic information for a cohort of patients with schizophrenia as well as a control group.

- **Readmission** Dua & Graff (2019c). This dataset contains 10 years of clinical care at 130 US hospitals for patients with diabetes. Each row represents a hospitalized patient records diagnosed with diabetes. The goal is to determine whether the patient would be readmitted to the hospital within 30 days of discharge.

## F Examples of Explanations Generated

See Figures 6 and 7. Note that examples are not shown for the Schizophrenia dataset as the gene names are not informative.

If at least 4 of the following conditions are satisfied:
- Gang Affiliated = 1
- Jobs Per Year = 0
- Arrests with PPV Violation >= 5
- Arrests on Property >= 2
- Age at Release $\leq$ 48

then predict 1 for recidivism.

a Recidivism: Support = 0.099, ABBR = 0.903

If at least 4 of the following conditions are satisfied:
- External Risk Estimate $\leq$ 63
- External Risk Estimate $\leq$ 53
- Trades with Balance $\geq$ 0.86
- Average Months in File $\leq$ 53
- Number of Total Trades $\leq$ 19

then predict 1 for FICO loan default.

b FICO: Support = 0.095, ABBR = 0.921

If at least 5 of the following conditions are satisfied:
- Education $\geq$ 13
- Age $\geq$ 33
- Relationship = Husband
- Work Class is not Missing
- Married = 1

then predict 1 for diabetes.

c Adults: Support = 0.102, ABBR = 0.930

If at least 4 of the following conditions are satisfied:
- Heart Disease = 1
- BMI $\geq$ 32
- High Cholesterol
- High Blood Pressure
- General Health $\geq$ 3

then predict 1 for Income $\geq 50K$.

d Diabetes: Support = 0.104, ABBR = 0.929

If at least 4 of the following conditions are satisfied:
- Admission Type = 6
- Number of Medications $\geq$ 9
- Admission Source $\geq$ 7
- Number Inpatietn $\geq$ 1
- Insulin = Yes

then predict 1 for readmission.

e Readmission: Support = 0.135, ABBR = 0.86

Figure 7: Examples of Checklists for $K = 5$

