# OpenReview forum: "Explaining Confident Black-Box Predictions"
_TMLR — Accepted by TMLR_

### Review · Reviewer_cvZ2 · 2025-04-08

**Summary Of Contributions:**

This paper proposes improvements to rule-based, model-agnostic explanation methods for binary classification model predictions.

1. While prior methods typically select rules that best explain binary (0/1) predictions, this paper selects rules and checklists that explain prediction patterns including confidence scores.

2. To quantify how well a rule explains model behavior, the paper introduces a new metric called Average Black-Box Ranking (ABBR).

3. To efficiently discover rules with high explanatory power, a local search-based optimization method is proposed.

4. The method is evaluated on 6 real-world datasets, and demonstrated that the algorithm reached high ABBR values compared to optimal possible ABBR.

**Audience:**

Yes

**Broader Impact Concerns:**

N/A.

**Claims And Evidence:**

No

**Requested Changes:**

**Critical to securing your recommendation:**

- Evaluation Setting:

    The evaluation should answer the following core question:

    > Given the same target support, do rules that have higher ABBR explain model behavior provide **better explanations**?

    For example, the paper proposes confidence-based explanations instead of explanations based solely on binary predictions. But is there a **quantitative experiment** showing that the former is better than the latter? There needs to be a **metric independent of ABBR** that can justify the superiority of ABBR-based explanations. There are two kinds of idea for evaluation benchmark:

    > [Idea 1] Suppose a method that simply maximizes support achieves support **x** on the train set and **y** on the test set. Now, if you use ABBR-maximizing rules (with target support **x**), does the resulting test set support **y′** become higher than **y**? If so, that could indicate the generalizability of ABBR-based rules.

    > [Idea 2] You could sample many random rules $\ell_k$ with the same support and show that **those with higher ABBR values tend to achieve higher test support**. That would empirically justify ABBR as a better criterion.

    Evaluation is always one of the most challenging yet crucial aspects of interpretability research. I would appreciate if the authors could come up with a compelling evaluation strategy.

---

**Would strengthen the work:**

- Theoretical justification for ABBR:

    Although I’m not certain, ABBR reminds me of **Mann–Whitney U statistic** / **Wilcoxon rank-sum test**, from non-parametric, rank-based statistical methods. Is there a precise mathematical concept that connects to ABBR?

    While ABBR is intuitively reasonable, the paper lacks a **formal justification for why it is a good metric for evaluating ranking quality**. The connection to consistency alone feels insufficient—especially since **consistency itself is not a strong enough metric**, which is the very motivation for introducing a ranking-based measure like ABBR.

- Figure 4: LaTeX rendering errors.

**Strengths And Weaknesses:**

**Strengths**

- Clear motivation:

    Confidence scores naturally carry rich information about model behavior and are highly relevant to concepts such as reliability, which is a key concern in explainable AI.

    Given a fixed support target, it is intuitive that explaining samples with higher confidence results in rules that better capture model behavior.

**Weaknesses**

- Evaluation:

    The paper aims to demonstrate the effectiveness of ABBR-based explanations. However, the experiments primarily focus on showing that high ABBR values can be achieved, rather than **providing evidence for why ABBR-based explanations are superior**.

    Table 2, the paper’s main empirical result, only demonstrates that high ABBR can be achieved. It does not empirically validate that achieving high ABBR leads to better explanations.

---

> ### Author Response · Authors · 2025-04-30
> **Response to Reviewer**
>
> _Thank you very much for your detailed review of our paper and the thoughtful suggestions. We respond to your feedback below._
>
> **Regarding the two experimental ideas suggested**, we very much appreciate the thought behind them and agree that they would be valuable to performm. We believe that our generalizability of our rules (when optimized for ABBR) will certainly be better than those based on the discrete 0/1 predictions. Our revised version of the paper will explore this in more detail.
>
> **Regarding the justifying the usefulness of the ABBR metric**, we elaborate some more on why finding rule lists with high ABBR for a given support is useful:
>
> Imagine you are trying to understand when a black-box model makes confident predictions. The underlying decision mechanism is inaccessible, but we want to find sufficient conditions — simple, interpretable patterns — that reliably describe when the model is confident. Our method, which optimizes for ABBR at a fixed support level, is designed to do exactly this: it produces statements like
> > "If any of rules $R_1, R_2, \ldots, R_M$ are satisfied, then the model's predicted probability is, on average, at the $p$-th percentile among all data points,"
> where $p$ is the ABBR.
>
> While we cannot fully explain the model’s behavior everywhere, we can identify interpretable subspaces — defined by a rule list — where the model consistently exhibits high confidence. The size of this interpretable subspace is the support. In this setting, ABBR is a natural metric—defining "high confidence" as having a high _average_ the percentile ranking induced by the black-box model. Other metrics are also possible, such as taking the minimum or the median. The reason we take the mean is simply because it’s the most natural, and because of the nice connection to average consistency against 0/1 predictions.
>
> We also want to highlight that this type of explanation we provide is novel in the literature. Please see the paragraph “Regarding the lack of comparison to state-of-the art baselines” to Reviewer qZvC, where we describe how our rule lists optimizing ABBR at a target support is a novel problem setting. We hope that this explanation sheds more light into why optimizing for ABBR at a given support level is a natural metric to consider.
>
> **Regarding the Mann-Whitney U statistic**, you are absolutely that there is a connection. The Mann-Whitney U statistic between two sets of points X of size $n$ and Y of size $m$ tests for whether the values of X are ranked more highly than those from Y. It is computed by taking all $n + m$ numbers, ranking them, computing the sum of the ranks of the points in X, same for Y, then applying some simple arithmetic.
>
> In our case, given a rule list that applies to the data points $L$, let $\bar{L}$ be the complement of $L$ (i.e., the data points that the rule doesn’t apply to). The Mann-Whiteney U-statistic between $L$ and $\bar{L}$ exactly involves computing the sum of the ranks of the points in $L$. Our revised version of our paper will fully explore this connection, use it as a justification for ABBR, and provide a full proof in the Appendix.

---

> > ### Comment · Reviewer_cvZ2 · 2025-05-05
> >
> > **[On the Usefulness of ABBR]**
> >
> > I agree with the intuitive explanation you provided regarding the usefulness of ABBR. However, as it stands, there doesn't seem to be any quantitative experiment in the paper that supports this claim. This isn’t to say that ABBR is invalid, but rather that its validity hasn’t been empirically verified.
> >
> > By the way, is it not possible to revise the paper or include additional experiments under TMLR? From what I could tell, there didn’t seem to be any new experiments related to this point in the current version of the paper. The reason I’m emphasizing this is that, from my perspective, there is currently no quantitative evidence demonstrating that ABBR leads to more generalizable rule-based explanations. This makes it difficult for me to judge the claim as sufficiently supported from a Claims and Evidence standpoint.
> >
> > **[On the Connection to the Mann-Whitney U Statistic]**
> >
> > Thank you for precisely formulating the connection. If I understand correctly, a higher ABBR means that the rule-based explanation $\ell$ offers a more accurate description of the conditions under which the model’s confidence is high. Understanding ABBR through a nonparametric statistical lens makes it feel like a much more grounded method to me.
> >
> > ---
> >
> > Given the current state of the paper, I will maintain my current rating.

---

> > > ### Comment · Action_Editor_Hhoc · 2025-06-04
> > >
> > > I would like to thank the authors and reviewers for engaging in the discussion.
> > >
> > > Dear authors,
> > >
> > > Similar to Reviewer cvZ2, I do not see any updated manuscript. TMLR allows you to modify the manuscript to incorporate reviewer feedback. Is it possible to include additional details for the two experimental suggestions pointed out by Reviewer cvZ2? And also add baselines to the paper as discussed in response to the Reviewers Yw4T and qZvC?

---

> > > > ### Author Response · Authors · 2025-06-04
> > > > **Response to Action Editor**
> > > >
> > > > Thank you for the comment. I did not realize we can edit the manuscript during the review process. I am in the process of adding the following experiments:
> > > >
> > > > * Idea 2 from Reviewer cvZ2, showing the value of ABBR through the fact that rules with high ABBR are more likely to generalize to the testing set.
> > > > * Baseline proposed in response to Reviewer Yw4T and qZvC involving gathering rules through Rudin and Shaposhnik.
> > > >
> > > > I anticipate that these experiments will highlight the value of ABBR as a metric and our approach broadly. I hope these will be sufficient in securing your recommendation. Thank you!

---

> > > > > ### Comment · Action_Editor_Hhoc · 2025-06-04
> > > > >
> > > > > That would be great, thank you!

---

> > > > > > ### Author Response · Authors · 2025-06-09
> > > > > > **Revision Posted**
> > > > > >
> > > > > > Dear Action Editor and Reviewer cvZ2,
> > > > > >
> > > > > > We have updated the manuscript with two revisions based on your feedback.
> > > > > >
> > > > > > (1) Empirical experiment to show the value of ABBR as a metric (Appendix A).
> > > > > > (2) Comparison of RuleListMiner algorithm against a suitable baseline (Appendix B).
> > > > > >
> > > > > > The paper body has been slightly modified with some text that refers the reader to Appendix A and B. These paragraphs are highlighted in blue text. Additionally, we have fixed typos pointed out by the other reviewers.
> > > > > >
> > > > > > Thank you!

---

> > > > > > > ### Comment · Action_Editor_Hhoc · 2025-06-09
> > > > > > >
> > > > > > > Thanks a lot for the quick update.
> > > > > > >
> > > > > > > Reviewer cvZ2, can you take a look, please?
> > > > > > >
> > > > > > > I will take a look at the full draft in the next few days as well.

---

### Review · Reviewer_Yw4T · 2025-04-14

**Summary Of Contributions:**

In this paper, the authors propose a novel metric and algorithm to improve model‑agnostic explanations of binary classifiers. The new metric, Average Black‑Box Ranking (ABBR), considers the confidence ranks (i.e., predicted probabilities) assigned by models to their predictions across a dataset. Based on this measure, two algorithms—RuleListSearch and ChecklistSearch—are designed to maximize ABBR while accounting for the sparsity of the explanation (in other words, interpretability). The authors demonstrate the effectiveness of their approach empirically across six datasets spanning healthcare, criminal justice, and finance, highlighting a clear trade‑off between interpretability and predictive accuracy.

**Audience:**

Yes

**Claims And Evidence:**

Yes

**Requested Changes:**

Typos and Formatting

Page 6, last paragraph: change “taking taking” to “taking.”

Page 10, Section 5, second paragraph, third line: change “he” to “The.”

Table 2 caption: replace "S∈{3,5}” with “𝑀∈{3,5}”

Table 3 caption: the caption is not visible and covered by a figure

Comparison with Alternative Methods
Include at least one baseline post‑hoc explanation technique (e.g., LIME or SHAP link to papers are above) in the experimental comparison.

Extension Beyond Binary Classification
Add a discussion of how ABBR—and the associated algorithms—could be adapted (or why they may not extend) to multi‑class classification and regression tasks. Highlight any theoretical (if possible) or practical obstacles.

Scalability With High‑Dimensional Data
The current datasets have 14–54 features. If feasible, run an additional experiment on a dataset with substantially more features (e.g., >100). If that is not possible, include a discussion of how the number of features affects candidate‑rule generation and explanation quality within your framework, and outline potential strategies to mitigate scalability issues.

**Strengths And Weaknesses:**

Strengths

Novelty and Relevance
Introduces the ABBR metric, an innovative approach for evaluating interpretability, and proposes a clear, practical method for directly optimizing interpretability metrics linked to model confidence.

Practical Impact
The developed algorithms are highly practical, easy to understand, and readily deployable. Demonstrated empirical validity across diverse, realistic datasets reinforces the usefulness of the approach for real‑world applications.

Methodological Clarity
Empirical experiments are thorough, with a clear presentation of comparative performance and optimality gaps.

Balance between Interpretability and Accuracy
The paper convincingly illustrates the trade‑offs between explanation sparsity (interpretability) and predictive performance (ABBR), enabling users to make informed decisions based on their needs.

Weaknesses

Scalability
Generating candidate rule sets may become computationally infeasible for extremely large or high‑dimensional datasets.

Comparison with Baselines
The approach focuses primarily on rule lists and checklists; the experiments lack comparisons with other post‑hoc explanation techniques such as LIME (https://dl.acm.org/doi/pdf/10.1145/2939672.2939778) or SHAP (https://dl.acm.org/doi/pdf/10.5555/3295222.3295230).

Generalization to Multi‑Class or Regression Tasks
The current method addresses only binary classification. Extending ABBR or similar metrics to multi‑class classification or regression remains an open question.

Typos
A few typographical errors remain (see “Requested Changes”).

---

> ### Author Response · Authors · 2025-04-30
> **Response to Reviewer**
>
> _Thank you for the detailed evaluation of our paper and for the suggested changes. We respond to your feedback below_
>
> **Regarding a comparison with alternative post-hoc explanation techniques like LIME or SHAP**, we would like to highlight the challenge in making a direct comparison between our approach and those approaches, but then propose a suitable benchmark based on a recent paper. LIME provides a local explanation for a single data point, whereas our rule lists provide a global explanation. LIME provides insight into how the features locally affect the model’s output, whereas our explanations provide global insights into sufficient conditions under which a model makes condition predictions. While SHAP is also a global explanation, it provides more of a feature importance map, whereas our rule list explanations provide explicit combinations of features that lead to a high predicted confidence.
>
> Please see our response to **Reviewer qZvC** above for a detailed response for why it is challenging to find a suitable benchmark in our problem setting. Nevertheless, we recognize the importance of having a benchmark to show the value of our approach. In our response, we propose a comparison a recent work by Rudin and Shaposhnik, one that we are confident will highlight the need to explicitly account for confidence when constructing rule lists.
>
> **Regarding extensions beyond binary classification**, we believe extensions to multi-class and regression can be done very naturally.
> * For multi-class classification, we believe the best way to apply our methodology is to do a one-versus-rest comparison. Asking: when does the model predict class A very confidently over classes B, C…etc.? We don’t think there is a natural way of extending ABBR in another way, since ABBR is based on a scalar ranking/probability.
> * For regression tasks, ABBR can be easily generalized by treating paralleling the predicted value of a regression model with the predicted probability of a classification model. We can rank the data points based on their predicted value of the regression model, then apply ABBR in the same way as before to explain when the model makes the highest/largest predictions.
>
> **Regarding scalability with high-dimensional data**, we believe our algorithms are fully scalable beyond the ~50 features we tested on. The crux of our RuleListSearch algorithm is the use of random forest to identify the candidate set of rules. Random Forest is able to run easily on datasets with high-dimension as it leverages heuristics to find a good split of each individual CART. Given a set of candidate rules, all of which are strong rules found from the leaf nodes of the random forest, our simulated annealing procedure can be run for as long or short as the user desires.

---

### Review · Reviewer_qZvC · 2025-04-20

**Summary Of Contributions:**

The paper introduces a new method for model-agnostic, rule-based explanations that focus on confident predictions of a black-box binary classifier. It replaces the traditional "explain the hard thresholded labels" view with an explanation objective defined over the ranking of predicted probabilities. Experiments on six datasets show it reaches small gap to the optimum.

**Audience:**

Yes

**Broader Impact Concerns:**

No concerns identified.

**Claims And Evidence:**

Yes

**Requested Changes:**

1. Add comparisons to state-of-the-art baselines.
2. Include comprehensive ablation studies: (i) Analyze the contribution of each key component (ii) Evaluate the sensitivity to important hyperparameters.
3. Provide thorough analysis of search algorithm choice.

**Strengths And Weaknesses:**

**Strengths**
1. The proposed method is efficient and scalable. The two-phase search runs efficiently and can produce rules on relatively large datasets.
2. Experiments on various datasets show promising performance.
3. The paper is well-written and easy to understand.

**Weaknesses**
1. The paper does not include comparisons with any baselines, making the empirical improvement unclear.
2. There is no ablation study to clearly demonstrate the effectiveness of each component and the effect of hyperparameters.
3. It would be beneficial to discuss why simulated annealing was adopted instead of other search algorithms such as Bayesian optimization and evolutionary computation.
4. Figure 4 overlaps with Table 3.

---

> ### Author Response · Authors · 2025-04-30
> **Response to Reviewer**
>
> _Thank you for the the thorough review of our paper. We appreciate you taking the time to evaluate and help us improve our work._
>
> **Regarding the lack of comparison to state-of-the art baselines**, we would like to highlight some challenges in choosing an appropriate state-of-the-art baseline, then propose a novel benchmark that we will compare to in our revision.
>
> Two key characteristics of our work the global explanations provided by rule lists (and checklists) and the ability to specify a desired support level (i.e. what proportion of the dataset we want to explain).
> * _Global Explanations_. Our constructed rule lists provide a sparse global explanation for when a black-box model is confident. Existing work that explain black-box models often focus on local explanations, which are explanations anchored on given data point. To explain why B(x) = 1, an if-then rule R is found which applies to x and optimizes a combination of support (applies to many other data points), and consistency (applies to many other data points where B(x) = 1). Although one can generate a rule for each data point, it is unclear how to combine those rules into a sparse list that explains the overall behavior of the model.
> * _Specifying a Desired Support Level_. Our methodology allows the user to specify a desired support level s and to find rules lists with higher ABBR that explain roughly s proportion of the population. Existing work involving global explanations of black-box models often try to find surrogate interpretable models that match the 0/1 predictions of black-box model, thereby providing a global explanation for when the model predicts 0 or 1. When trying to explain when a model makes confident predictions, optimizing for accuracy is insufficient--a rule list may replicate a black-box model’s 0/1 predictions with high accuracy — for example, 90% — but this does not tell us when the model is confident. Accuracy-based evaluation favor explanations that cover large parts of the dataset, regardless of prediction confidence.
>
> Given these differences, there is no existing method that provides a direct comparison. That being said, we recognize the importance of having a benchmark, so we propose a benchmark based on the paper by Rudin and Shaposhnik (2023) titled _Globally Consistent Rule-Based Summary-Explanations_ published in JMLR. This paper finds if-then rules that provide local explanations to a given data point. We propose the following benchmark:
> 1. Generate a candidate set of rules by applying Rudin and Shaposhnik to every data point where the black-box model predicted 1. This can be done with a cutoff probability of 0.5.
> 2. Leverage our `RuleListSearch` algorithm with simulated annealing on this candidate set of rules.
> This would provide a meaningful benchmark for an apples-to-apples comparison. We believe our methodology will be superior due to the fact that our candidate set of rules is optimized for ones with high ABBR, rather than just consistency with the 0/1 predictions.
>
> **Regarding the lack of ablation studies**, we agree that more ablation studies could be performed, but want to highlight a few ablations that our work did investigate:
> * $L$ – the maximum number of conditions allowed in any rule
> * $M$ – the maximum number of rules allowed in a rule list.
>
> Our main results (Table 2) shows that increasing $L$ and $M$ result in rule lists with better ABBR. The increase in $L$ is particularly interesting because it shows that Phase 1 of our RuleListSearch algorithm is crucial. Phase 1 generates the candidate set of rules, which are then selected to form the rule list. Table 2 shows that in order to achieve rules with high ABBR, a large $L$ is needed (i.e., rules that involve the AND of many conditions, see Table 3). While for $L = 2$ an exhaustive enumeration of all rules is possible, this is not feasible for $L > 2$ due to the combinatorial explosion of possible rules. Hence, a heuristic-based approach to generate candidate rules is necessary, thus motivating the use of Random Forest approach.
>
> **Regarding the use of simulated annealing as our search algorithm of choice**, our work proposes it as a standard optimization technique in this discrete space, but admittedly we didn’t find it in the scope of this initial paper to investigate the optimization procedure itself. It would be a fruitful direction for follow-up research to see if we there are any optimization contributions that can be made to. Optimizing for ABBR is fundamentally challenging due to the discrete nature of the problem: when constructing a rule list of size $M$ out of a candidate pool of size $N$, we are essentially optimizing over $N$ choose $M$ possible rule lists, a very discrete space.

---

### Decision · Action_Editor_Hhoc · 2025-06-27

**Recommendation:** Accept with minor revision

**Additional Comments:**

I would suggest incorporating the newly added Appendices A and B into the main body of the paper. I believe these experiments are necessary to showcase the effectiveness of your metric. I also suggest proofreading the paper for typos (e.g., a space missing before Gap in the second-to-last paragraph on page 16).

**Audience:**

Yes

**Audience Explanation:**

The topic of the paper and the provided approach will be of interest to a broad set of TMLR audience.

**Claims And Evidence:**

Yes

**Claims Explanation:**

This paper proposes improvements to rule-based, model-agnostic explanations for binary classification by taking into account the confidence of the model when making predictions. The authors propose a new metric, ABBR, that measures the number of data points for which the model has confident predictions. The authors propose a local search-based optimization method to discover rules with high ABBR.

The authors justify the usefulness of ABBR and demonstrate the effectiveness of the search-based optimization approach on 6 datasets.